# Data Imputation of Soil Pressure on Shield Tunnel Lining Based on Random Forest Model

**DOI:** 10.3390/s24051560

**Published:** 2024-02-28

**Authors:** Min Wang, Xiao-Wei Ye, Xin-Hong Ying, Jin-Dian Jia, Yang Ding, Di Zhang, Feng Sun

**Affiliations:** 1Polytechnic Institute, Zhejiang University, Hangzhou 310058, China; 11912133@zju.edu.cn; 2Department of Civil Engineering, Zhejiang University, Hangzhou 310058, China; cexwye@zju.edu.cn (X.-W.Y.); 22212018@zju.edu.cn (X.-H.Y.); 22212019@zju.edu.cn (J.-D.J.); 3Department of Civil Engineering, Hangzhou City University, Hangzhou 310015, China; 4China Railway Siyuan Survey and Design Group Co., Ltd., Wuhan 430063, China; zdiok@126.com (D.Z.); sunfeng5433@126.com (F.S.)

**Keywords:** shield tunnel, soil pressure, missing data, imputation, random forest

## Abstract

With the advancement of engineering techniques, underground shield tunneling projects have also started incorporating emerging technologies to monitor the forces and displacements during the construction and operation phases of shield tunnels. Monitoring devices installed on the tunnel segment components generate a large amount of data. However, due to various factors, data may be missing. Hence, the completion of the incomplete data is imperative to ensure the utmost safety of the engineering project. In this research, a missing data imputation technique utilizing Random Forest (RF) is introduced. The optimal combination of the number of decision trees, maximum depth, and number of features in the RF is determined by minimizing the Mean Squared Error (MSE). Subsequently, complete soil pressure data are artificially manipulated to create incomplete datasets with missing rates of 20%, 40%, and 60%. A comparative analysis of the imputation results using three methods—median, mean, and RF—reveals that this proposed method has the smallest imputation error. As the missing rate increases, the mean squared error of the Random Forest method and the other two methods also increases, with a maximum difference of about 70%. This indicates that the random forest method is suitable for imputing monitoring data.

## 1. Introduction

In the present era, as computer science and technology continue to progress, underground shield tunneling projects have embraced cutting-edge technologies like artificial intelligence and big data. This integration facilitates the seamless acquisition and monitoring of comprehensive lifecycle data for shield tunnel segments. This is aimed at studying the deformation mechanisms and mechanical performance evolution of tunnel segments [1,2,3,4,5]. The tunnel segment monitoring system acquires force and displacement data of the soil, segments, and components by installing buried sensors and surface-mounted sensors at key locations. The data are then uploaded to a cloud platform in real time through remote communication. Analyzing all the data allows for the assessment of the safety of the tunnel structure. Should critical data be lost, leading to data gaps, the accuracy of the tunnel structure safety assessment becomes compromised, thereby impeding the timely avoidance of potential engineering hazards. Therefore, handling data gaps is of paramount importance. During this process, despite the meticulous protection measures implemented for data collection and transmission equipment, data gaps remain widespread. Data gaps are prevalent to varying degrees in various data sources [6,7,8,9].

At present, many scholars from countries such as China, the United States, the United Kingdom, Switzerland, Canada, and Germany have conducted research on data missingness handling methods in various fields such as healthcare, economy, and transportation. According to the research by Little and Rubin, there are three mechanisms for data missingness, including Missing Completely at Random (MCAR), Missing at Random (MAR), and Missing Not at Random (MNAR) [10,11]. In the case of MCAR, the missing data are unrelated to any factors and do not follow any pattern. In the MAR case, the missing data are related to certain characteristics and variables of the existing data but not directly related to the missing values themselves. In the MNAR case, the missing data are not only related to the existing data but also related to unknown observed variables.

In previous research, common methods for handling data missingness included Listwise Deletion (LD) or Pairwise Deletion (PD), both of which belong to deletion methods that involve directly removing the missing data from the dataset. Although PD, based on LD, considers each variable separately and reduces the number of cases deleted, both methods result in the loss of useful information due to the researcher’s selection and significantly reduce the sample size. When the data missingness is small, such as below 5%, and the missingness mechanism is MCAR or MAR, direct deletion methods can be appropriate. However, when the data missingness is large or the missingness mechanism is MNAR, simply using deletion methods can lead to substantial estimation bias [12,13,14,15,16].

In subsequent research, to address the issue of loss of useful information caused by deletion methods, imputation methods began to be used to handle incomplete data. Imputation is the process of utilizing estimated values to fill the gaps in incomplete data. In general, the two main approaches to imputing missing values include employing statistical techniques or utilizing machine learning techniques to estimate and fill in the missing data.

The median imputation method involves substituting missing values with the median value derived from the available data for that specific feature. It is suitable for numerical data, particularly in the presence of outliers in the dataset. However, median imputation fails to utilize information from other related variables, leading to potentially substantial bias in the imputed values. Furthermore, it does not account for data uncertainty and variability [17,18,19]. Mean imputation involves computing the mean of a variable from cases that contain data and replacing the missing values for that variable with this mean value [20]. Although this technique can reduce variance, it ignores the correlation between variables and is appropriate only for MCAR or MAR missing data mechanisms and small sample sizes with only a few missing data points [21]. EM algorithm is an iterative optimization algorithm used for parameter estimation and often employed for unsupervised learning problems. It typically converges to a local optimum in parameter estimation but may face computational challenges for large-scale data due to the high computational cost [22,23]. Linear regression is a technique for analyzing various types of data, including continuous, categorical, and binary data. However, for nonlinear and complex relationships, traditional linear regression algorithms may not perform well [24,25,26]. K-Nearest Neighbors Imputation (KNN) is a technique that estimates missing values by identifying the K nearest neighboring samples from the available data. It considers the interrelationships between features and leverages information from other variables to fill in the missing values. However, it has several notable drawbacks. For large-scale datasets, the computation of distances between samples can be extremely time-consuming. Moreover, the selection of an unsuitable K value can result in biased imputation values. Additionally, KNN imputation may perform suboptimally when confronted with high-dimensional feature spaces due to the inherent challenges associated with the curse of dimensionality [27,28]. Multiple Imputation (MI) is a method that utilizes Monte Carlo simulation to generate multiple plausible complete datasets and derives the final imputed values through averaging or combining regression estimates. It enables simultaneous estimation of multiple missing values while accounting for uncertainty and variability, thereby enhancing result accuracy. However, MI entails a complex process involving multiple model fits and imputation operations. The computational cost is high. Additionally, it requires careful consideration of feature interdependencies and assumptions about the missing data mechanism [29,30].

The actual monitoring data of tunnel segments exhibit time correlation. In comparison to other time series datasets, sensor time series datasets typically feature high frequency, multiple variables, and long duration, while also being susceptible to noise interference from sensor errors and external factors. This article proposes using the Random Forest method to complete the missing parts in the dataset. It is a precursor to the classification tree algorithm and has now been driven by the development of ensemble learning and decision tree algorithms. The Random Forest algorithm has higher accuracy than most individual algorithms, rapid training speed, and can handle large-scale high-dimensional data. At the same time, each tree can be independently and synchronously generated, making it easy to parallelize. Therefore, it is widely used in various research fields. Section 2 of this article provides a brief introduction of the engineering background. Section 3 introduces the tunnel liner monitoring system and the possible causes of missing data. Section 4 explains the principle of the Random Forest algorithm. Section 5 describes the experimental steps and compares the results of imputing missing data using the median, mean, and Random Forest algorithms. Section 6 summarizes the work of this study and presents conclusions.

## 2. Engineering Background

Tunnel boring machines have the advantages of low cost, minimal environmental impact, fast construction speed, and minimal interference from seasonal weather, and have been widely used in subway, railway, and river transport. The tunnel project in this paper is located 3 km downstream of Fuyang Bridge, and uses a shield machine with a diameter of 15 m or more. The excavated soil layer is mainly composed of gravel and round gravel, and the bottom of the excavation section is fully weathered quartz diorite with a strength of about 40 MPa. The ratio of excavated gravel soil is high, with a particle size generally between 30–80 mm (accounting for 60–70%), and is shown in Figure 1. The tunnel passes through strata with high permeability, and the overlying soil layer during construction is thin, with a minimum cover thickness of less than 0.7D. Therefore, during the period when the segment is assembled into a ring to resist the pressure of the strata along with the grout final setting and the segment, the segment will be more likely to float due to various factors such as the pressure of the shield machine hydraulic jacks, injection pressure, and ground reaction forces, which can lead to segment damage. On the other hand, the tunnel is a highway-rail joint venture, and during operation, complex traffic dynamic loads (such as subway train loads and vehicle loads) and changes in water level will change the distribution of internal and external loads of the tunnel structure, causing uneven longitudinal settlement of the load structure, and longitudinal uneven deformation of the tunnel structure. This can cause stress concentration in weak parts of the tunnel structure, leading to cracks and water leakage in the segments, affecting service safety. The traditional operation and maintenance methods have encountered significant challenges in efficient diagnosis of structural damage and accurate assessment of operational performance. Engineering accidents caused by untimely maintenance occur frequently. Therefore, to address the key technical issues of deformation mechanism and mechanical performance evolution throughout the entire life cycle of tunnel boring machine (TBM) segments, this project needs to solve the following two technical challenges:Evaluation of Deformation and Initial Internal Forces for Shield Tunnel Structures: During the construction phase, a shield tunnel structure experiences upward shape deformation, resulting in the generation of initial internal forces in the segment and bolts. Precisely understanding the changes in initial internal forces during the construction phase for a shield tunnel structure is the fundamental basis for assessing the mechanical performance of the structure during the operation phase. Currently, shield tunnel segment design methods mainly include the traditional method, modified traditional method, and multi-hinge ring method. These methods primarily focus on analyzing the stress of segments during the regular service period and do not consider the impact that different factors may have on segment stress during the construction phase. Significant experience in shield tunnel engineering has shown that it takes some time for the segments to be assembled into rings and for the grout to solidify, with joint resistance and external ground pressure affecting the load transfer mechanism. Thus, it is essential to systematically study the mechanical behavior of segments considering the soil–segment–jack interactions and their key influencing factors. Furthermore, analyzing the ground disturbance caused by the shield excavation and tunnel additional settlement can accurately evaluate the initial stress state of the tunnel structure.Long-term Deformation and Mechanical Performance Evolution of Shield Tunnel Structures During the Operational Phase: The water level of the Fuchun River fluctuates seasonally, causing changes in the external confining pressure on the tunnel and affecting the deformation and internal forces of the tunnel structure. This can result in water leakage through the segments, compromising the overall stability of the tunnel structure and causing damage to electrical equipment inside the tunnel. Moreover, localized water leakage can lead to weathering and detachment of the concrete lining. Corrosive substances present in the water can accelerate the deterioration of reinforced concrete structures, reducing their load-bearing capacity. Furthermore, during the operational phase, complex traffic loads significantly influence the mechanical performance of the segments. Interactions between vehicles, trains, and the road surface or tracks cause vibrations in the shield tunnel structure, which are transmitted to the surrounding soil. This process can cause plastic deformation and result in the buildup and release of pore pressure within the soil. Consequently, both the tunnel structure and the surrounding soil undergo uneven settlement and deformation. In addition to the natural degradation of the structure over time, cumulative damage from long-term traffic loads can eventually lead to cracking, leakage, and misalignment of the tunnel structure. Moreover, shield tunnel structures constructed in combination with public transport systems feature complex internal structures such as roadway slabs and sidewalls. Traditional research methods that simplify these structures into single circular rings are unable to accurately analyze the dynamic response characteristics and deformation patterns of large-section tunnel structures with complex internal spatial divisions. Therefore, there is an urgent need to study the long-term deformation mechanisms and mechanical performance evolution of segments under the combined effects of water level fluctuations and traffic loads.
Figure 1Geological profile map.
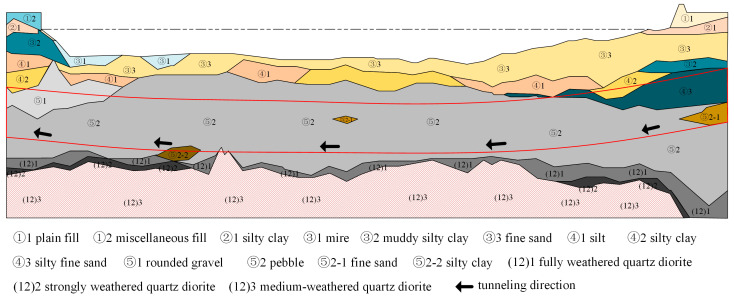


In summary, in view of the different stress characteristics of the segment structure of the tunnel throughout its entire lifespan, it is necessary to comprehensively analyze the deformation and internal force monitoring data of the shield tunnel structure during the construction phase. This will facilitate a holistic comprehension of the deformation and internal forces acting on the tunnel structure, along with variations in external loads. Furthermore, it allows for an analysis of the stress conditions experienced by the segment during the construction phase. Based on the determination of the mechanical behavior and initial internal forces of the segment during the construction phase, it is important to study the long-term longitudinal deformation characteristics of the tunnel structure under the combined effects of fluctuating river water levels and complex traffic loads. This will reveal the deformation mechanism and mechanical performance evolution of the shield tunnel segments throughout their entire lifespan, and propose deformation control methods and key indicators for the tunnel structure. This research has significant implications for the design optimization and safe operation of large-diameter river-crossing tunnels.

## 3. Monitoring System

The monitoring system consists of three subsystems, including the sensor subsystem, the data acquisition and transmission subsystem, and the cloud platform data management subsystem. Figure 2 shows the types of installed sensors and data acquisition and transmission instruments, where Figure 2a shows the surface-mounted sensors, including static water level gauges and vibrating wire strain gauges; Figure 2b,c show the embedded sensors, including earth pressure cells, steel strain gauges, concrete strain gauges, thermometers, and seepage gauges; and Figure 2d shows the data acquisition and transmission equipment. During the construction and operation phases, the stress and displacement data of the tunnel structure collected by the sensors are saved in real-time to the cloud platform for unified management through the data acquisition and transmission subsystem.

Considering that the soil pressure around the tunnel lining is the main load sustained during tunnel shield construction and operation, monitoring data of soil pressure can provide information on stress distribution in the surrounding soil and potential instability, which has a significant impact on the stability and safety of the tunnel. Moreover, analyzing soil pressure monitoring data allows for identifying deformation characteristics of the tunnel lining and its relationship with mechanical performance and soil pressure, aiming to optimize tunnel design and construction schemes in the future. This article adopts soil pressure time series data. The data are collected by pressure sensors installed on the outside of the lining segments. Each lining segment has an outer diameter of 15.2 m, thickness of 0.65 m, and width of 0.65 m. The segments are assembled with 1/3 overlap, consisting of seven standard segments (B), two connection segments (L), and one top segment (F). Each small segment has monitoring points, as shown in Figure 3 and Figure 4. The data for the entire process are collected using fiber-optic grating sensors, which offer advantages such as high precision, strong anti-interference capability, compact size, and fast response compared to other monitoring methods. They are more suitable for long-term monitoring of tunnel structures [31,32]. The equation for calculating soil pressure is as follows:(1)$$P=\frac{\lambda -\lambda_{0}-\left(\lambda_{t}-{\;\lambda }_{t}^{\prime }\right)\times \frac{1}{a}\times b}{K},$$

In the above equation, P is the soil pressure λ and $\lambda_{0}$ represent the measured and initial wavelengths, respectively; $\lambda_{t}$ and $\lambda_{t}^{\prime }$ represent the temperature compensation measured and initial wavelengths, respectively; a is the temperature compensation sensitivity coefficient; b is the temperature coefficient; and K is the coefficient of the primary term for soil pressure.

Due to suboptimal installation and working conditions of the sensors, there are several factors that can lead to data loss, such as (1) damage to some sensors during the process of tunnel lining pouring, maintenance, and transportation; (2) as the length of the shield tunnel reaches 1258 m, the probability of fiber optic damage increases with the increase in length of the fiber optic connection line during data collection and transportation; (3) during the construction phase, data collection interruptions may occur due to power outages, mechanical failures, or other human factors, resulting in data loss. Therefore, it is necessary to utilize existing data and suitable algorithms to achieve data recovery.
Figure 3Sensor distribution profile chart.
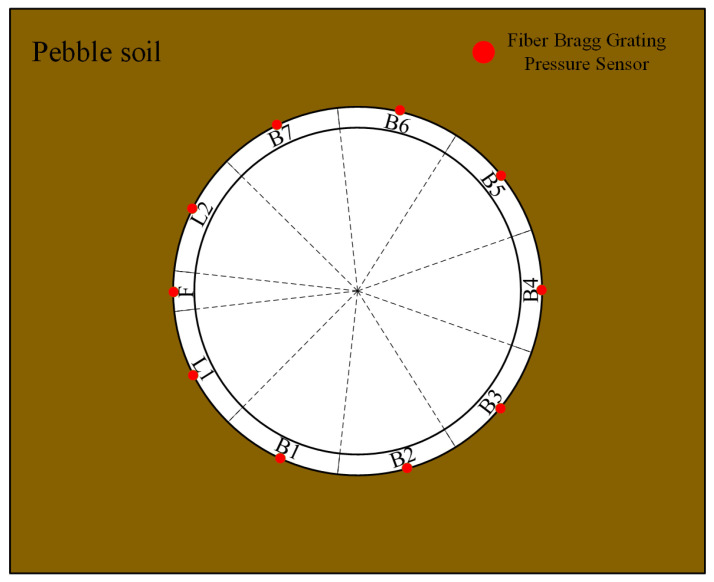

Figure 4Sensor distribution planar chart.
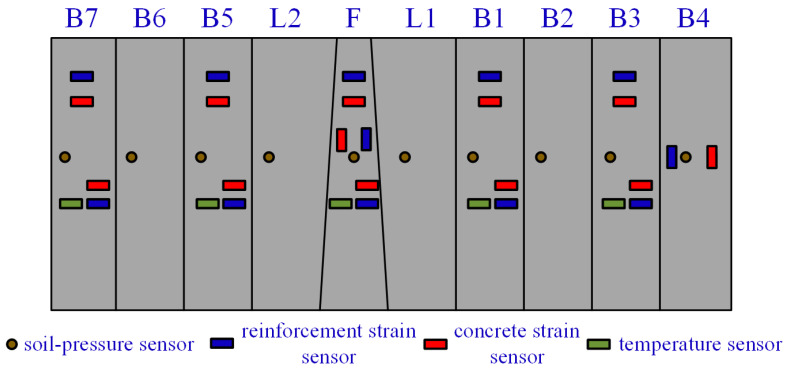


## 4. Random Forest (RF) Introduction

### 4.1. Development History

In the 1980s, Leo Breiman and others invented the classification tree algorithm, which greatly reduced the computational complexity by recursively partitioning the data for classification or regression. In 2001, Breiman [33] introduced Random Forest (RF) by combining classification trees. RF employs randomization techniques to utilize variables (columns) and data (rows), generating numerous classification trees and then aggregating their outcomes. In the following decades, scholars and experts from various fields carried out further theoretical analysis and experimental validation. For example, Andy Liaw and Matthew Wiener introduced the specific implementation methods of Random Forest in classification and regression problems and provided examples of using the Random Forest algorithm in the R language. Cutler et al. explored the application of the Random Forest algorithm in the field of ecology and demonstrated its practical effectiveness in ecological data classification through specific case studies [34,35]. Today, the development of the Random Forest algorithm has been driven by the advancement of ensemble learning and decision tree algorithms. Its practical applications have seen widespread adoption, propelling it to emerge as a crucial algorithm in the realm of machine learning over time.

### 4.2. Principle of Random Forest

Random Forest, classified as an ensemble learning method within the Bagging category, leverages the aggregation of numerous weak classifiers. Through voting or averaging techniques, the final outcome is obtained, yielding an overall model with superior accuracy and generalization capabilities. Bootstrap sampling is employed in Bagging to randomly select a predetermined number of samples from the training set. After each sample is selected, it is put back, and the number of samples collected is generally less than the original sample size. In this way, k new datasets are selected to train the classifier. During each round of random sampling in Bagging, certain data points from the training set are left out, forming what is known as Out of Bag (OOB) data. These data points are not used in training the model and can be utilized to assess the model’s capability to generalize to unseen instances. For classification problems, a simple voting method is usually used to obtain the class with the highest number of votes or one of the classes as the final output of the model. For regression problems, the final output of the model is typically obtained using a simple averaging method. This involves taking the arithmetic mean of the regression results generated by T weak learners. The Bagging structure is shown in Figure 5.

The Random Forest algorithm employs the CART decision tree as its weak classifier, which stands for Classification and Regression Tree. In cases where the dataset’s dependent variable is continuous, the decision tree functions as a regression tree. This means that the prediction value is determined by the average value observed at the leaf node. When the dependent variable is categorical, the decision tree acts as a classification tree, which effectively solves classification problems. This algorithm is a binary tree, meaning each non-leaf node can only create two branches. Therefore, when a non-leaf node represents a multi-level (more than 2) discrete variable, the variable may be referenced multiple times. Additionally, if a non-leaf node represents a continuous variable, the decision tree treats it as a discrete variable during the processing.

The CART decision tree relies on the *Gini* coefficient (GINI) for selecting features. The GINI criterion aims to maximize purity within each child node, ensuring that all observations within a child node belong to the same class. By minimizing the *GINI* coefficient, we can maximize purity and reduce uncertainty. In a decision tree with *K* classes, if the probability of a sample belonging to the *K*^th^ class is *P_K_*, the *Gini* index of this probability distribution can be calculated as follows:(2)$$Gini\left(p\right)=\sum_{K=1}^{K}p_{K}\left(1-p_{K}\right)=1-\sum_{K=1}^{K}p_{K}^{2},$$

In this equation, $Gini\left(p\right)$ stands for the Gini coefficient, where p represents the probabilities of different categories, and K denotes the number of categories. A larger value of *Gini*(*p*) indicates higher uncertainty, while a smaller value of *Gini*(*p*) indicates lower uncertainty and more refined data segmentation. Since the CART decision tree is a binary tree, it can be represented using the following formula:(3)$$Gini\left(p\right)=2p(1-p),$$

When exploring each feature and split point, if we divide the dataset *D* into two parts, *D*_1_ (containing samples with feature *A* = *a*) and *D*_2_ (containing samples without feature *A* = *a*), we can calculate the *Gini* of *D* with respect to feature *A* = *a* as follows:(4)$$Gini\left(D,A\right)=\frac{\left|D_{1}\right|}{\left|D\right|}Gini\left(D_{1}\right)+\frac{\left|D_{2}\right|}{\left|D\right|}Gini\left(D_{2}\right),$$

In this context, D refers to the original dataset, A represents the feature to be partitioned, $D_{1}$ denotes the set of samples that satisfy *A* = *a*, and $D_{2}$ represents the set of samples that do not satisfy *A* = *a*. $Gini\left(D\right)$ represents the uncertainty of set D, while $Gini\left(D,A\right)$ represents the uncertainty of set D after it has been partitioned based on *A* = *a*.

Random Forest utilizes multiple CART decision trees, where each tree is constructed by iteratively exploring all potential split points within the subset of features. It identifies the split point of the feature with the lowest Gini index to divide the dataset into two subsets, repeating this process until a stopping condition is satisfied. The concept of Random Forest is depicted in Figure 6.

### 4.3. Advantages of Random Forest

The translation of the given text is as follows:Due to the adoption of an ensemble algorithm, Random Forest itself has higher accuracy than most individual algorithms;It performs well on the test set. The introduction of two random elements makes Random Forest less prone to overfitting and provides a certain level of noise resistance, giving it an advantage over other algorithms;When trees are combined in Random Forest, it can accommodate nonlinear data and itself represents a nonlinear classification (fitting) model;It can handle high-dimensional data without the need for feature selection and show robustness to the dataset. Furthermore, it can handle both discrete and continuous inputs without normalization of the data;Owing to its Out-of-Bag (OOB) error estimate, it can obtain an unbiased assessment of the true error during the model building process without discarding any training data. During training, Random Forest can identify interactions amid features and determine each feature’s significance, thereby providing a valuable reference;As each tree within Random Forest is generated independently and concurrently, it is easy to parallelize the process, and it demonstrates fast training speeds to fit large-scale datasets.

## 5. Experiment

### 5.1. Introduction to Measured/Tested Data

In order to evaluate the interpolation effect of the Random Forest (RF) method, it is necessary to select samples with sufficient and complete data. Therefore, the raw data for this experiment are collected from fiber optic grating pressure sensors on eight sections of a certain ring (F, L1, L2, B2, B4, B5, B6, B7). Considering the requirement for an adequate number of samples and continuity, this experiment uses 1 h soil pressure data with a time interval of 1 s for data acquisition. As a result, the length of the time series for each monitoring point is n = 3600. The original soil pressure monitoring data are shown in Figure 7.

This study classifies the missing types as random missing (discontinuous data missing) and uses a complete dataset without missing data. Therefore, three missing datasets with missing ratios of 20%, 40%, and 60% are artificially created. Taking a 20% missing rate as an example, with a total of 28,000 data samples, we created an array during the artificial missing data generation process. This array consists of 5760 column indices distributed between 0 and 7, and 5760 row indices distributed between 0 and 3599. Afterwards, we assigned null values to the 5760 positions in the dataset based on the generated indices within the specified ranges. The curves of different missing ratios are shown in Figure 8, Figure 9 and Figure 10.

### 5.2. Random Forest-Based Imputation Method

Any regression learns from the feature matrix and solves for the continuous label. The reason regression algorithms can achieve this is that they believe there is a connection between the feature matrix and the label. In fact, the label and the features can be transformed into each other, and the idea of using regression to fill in missing values utilizes this.

For a dataset with n features, if feature T contains missing values, we consider feature T as the label and form a new feature matrix with the remaining n − 1 features and the original label. The non-missing part of feature T contains both labels and features, while the missing part only has features without labels. This missing portion needs to be predicted.

When there are missing values in features other than feature T, the Random Forest algorithm examines all the features and begins by filling in the missing values of the feature with the fewest missing values (since this requires the least amount of accurate information). When imputing a feature, the missing values of other features are substituted with 0. After each regression prediction is made, the predicted value is inserted back into the original feature matrix, and the process continues with the next feature. Each time the filling is complete, the number of features with missing values decreases, so the number of features that need to be filled with 0 decreases after each iteration. Once the algorithm reaches the final feature, which typically has the highest number of missing values among all the features, there are no other features remaining to be filled with 0.

### 5.3. Comparison of Imputation Methods and Evaluation Metrics

This study compares the proposed imputation method with two other commonly utilized techniques for imputation, specifically, mean imputation and median imputation. The evaluation method used is mean squared error (MSE), which is calculated as follows:(5)$$MSE=\frac{1}{m}{\sum_{i=1}^{m}\left(y_{test}^{\left(i\right)}-{\hat{y}}_{test}^{\left(i\right)}\right)}^{2},$$

In this case, m refers to the number of samples, $y_{test}^{\left(i\right)}$ represents the true values of the test set, and ${\hat{y}}_{test}^{\left(i\right)}$ represents the predicted values of the test set. MSE calculation is simple and can directly reflect the overall performance of the prediction model. The smaller the value of MSE, the more accurate the prediction model.

### 5.4. Parameter Selection and Optimization

In Random Forest-based imputation methods, several parameters are typically optimized, including (1) the number of decision trees, which can improve the performance and stability of the Random Forest but may increase computation time and require an optimal value; (2) the maximum depth of each decision tree to prevent underfitting or overfitting; (3) the number of features considered for each node split; and (4) the minimum number of samples required to be in a leaf node to avoid overfitting due to too few samples.

In this study, cross-validation was used to select the optimal parameters. The commonly used numbers of cross-validation folds are 5-fold cross-validation and 10-fold cross-validation. We chose 5-fold cross-validation in this study because it requires fewer computational resources and helps to balance the trade-off between variance and bias when evaluating model performance. Firstly, the dataset was divided into 5 subsets, and then the Random Forest regressor was trained and evaluated. The evaluation metric used is mean squared error to find the minimized objective function. ntree = 50, 150, 200, Max depth = 2, 4, Max features = 2, 4, 8 are iteratively run for 18 parameter combinations, resulting in model evaluation metrics as shown in Figure 11, Figure 12 and Figure 13.

Since the model evaluation is based on mean squared error (MSE), lower scores indicate better performance. From the results, it can be observed that the model performs better when Max depth = 4, and the number of features has a less significant impact on the model. Subsequently, we select ntree = 50, 100, 200 and Max depth = 4, Max features = 4 for comparison, as shown in Figure 14.

### 5.5. Imputation Results and Comprehensive Evaluation

We applied the aforementioned parameters to impute missing data in datasets with missing rates of 20%, 40%, and 60% for soil pressure. Figure 15, Figure 16, Figure 17 and Figure 18 demonstrate the comparison between the results obtained using different imputation methods (median, mean, and Random Forest) and the original data for four segments (F, L1, L2, B1). It can be observed that the imputation results of median and mean methods are poor, with a significant difference between the imputed values and the original values. On the other hand, the Random Forest imputation method produces good results with a high degree of agreement between the imputed values and the original values.

The comparison of the mean squared error (MSE), which is used as the evaluation criterion for assessing the performance of the model in imputing missing data, is presented in Figure 19, Figure 20 and Figure 21. From the comparison, it can be inferred that there is a positive correlation between the missing rate and the imputation error. As the missing rate increases, the error of the Random Forest-based imputation method remains around 0.000250, indicating its good robustness. Across different missing rates, the Random Forest-based imputation method consistently has the lowest error. Particularly, at a missing rate of 60%, the Random Forest-based imputation method achieves the smallest error of 0.000249, which is approximately 70% lower than the errors of other methods.

## 6. Conclusions

This paper mainly accomplishes the following three tasks: firstly, the behavior of tunnel segments is monitored through both surface-mounted and embedded sensors, characterizing the stress and displacement changes throughout the entire lifespan of the segment structure; secondly, a Random Forest-based interpolation method is developed, and by comparing the MSE scores, the optimal combination of the number of decision trees, maximum tree depth, and maximum number of features is determined; thirdly, incomplete soil pressure datasets with missing proportions of 20%, 40%, and 60% are completed using data imputation, and the imputation results are compared with models using median and mean imputation methods. Based on the above work, the following conclusions are drawn:According to the field monitoring results, it is evident that the soil pressure exhibits minimal fluctuations in the first 25 min, followed by a gradual decline, indicating a non-linear variation;The Random Forest model demonstrates optimal performance and achieves the minimum mean squared error (MSE) when the following parameters are set: 200 decision trees, a maximum depth of 4 for each tree, and the consideration of a maximum of four features during each node split;As the missing proportion increases, the imputation errors of the models based on median and mean imputation methods also increase, while the error of the model based on Random Forest remains around 0.00025. It is evident that the Random Forest method outperforms median and mean imputation methods. At a missing proportion of 60%, the difference in errors reaches approximately 70%;Comparing the interpolated results with the original data through plots shows that the Random Forest-based imputation method can effectively handle multidimensional data obtained from sensor monitoring. It can provide reasonable predictions to fill in the missing parts of the dataset.

## Figures and Tables

**Figure 2 sensors-24-01560-f002:**
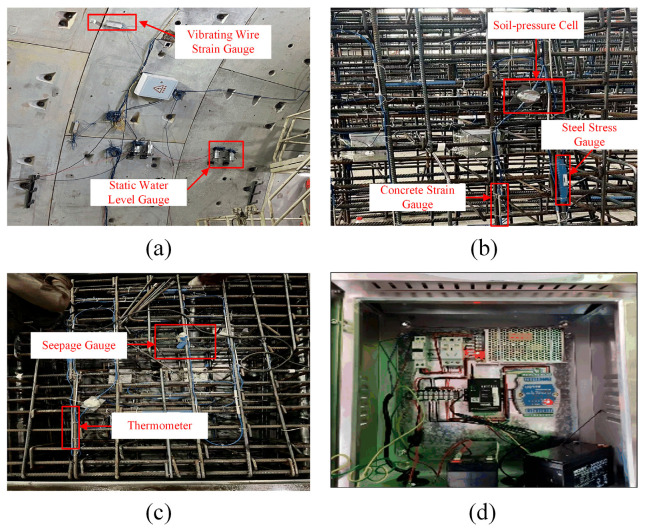
Sensor schematic diagram. (**a**) surface-mounted sensors; (**b**,**c**) embedded sensors; (**d**) data acquisition and transmission equipment.

**Figure 5 sensors-24-01560-f005:**
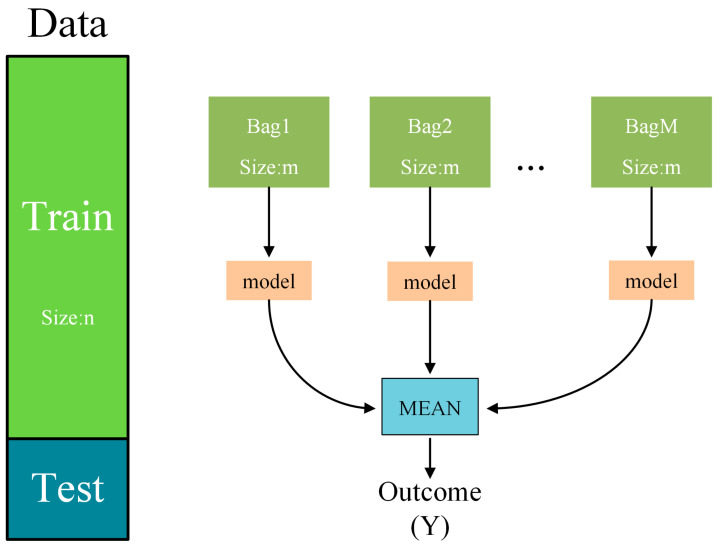
Bagging principle diagram.

**Figure 6 sensors-24-01560-f006:**
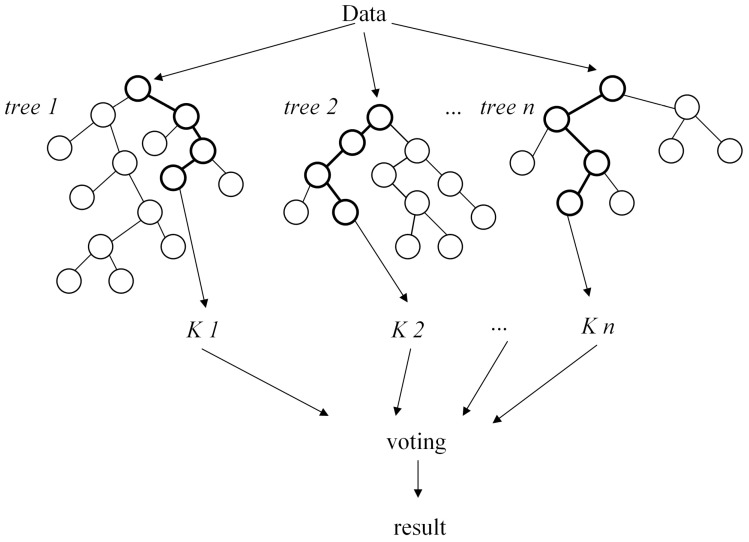
Random Forest principle diagram.

**Figure 7 sensors-24-01560-f007:**
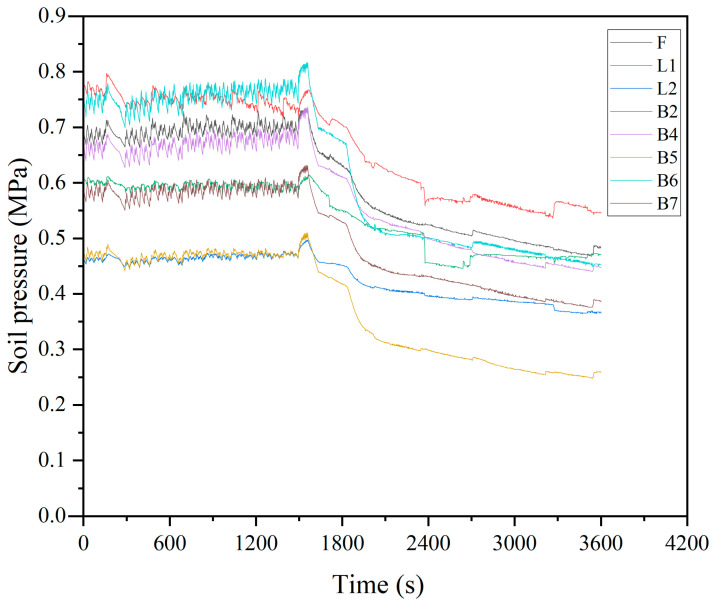
Original data plot.

**Figure 8 sensors-24-01560-f008:**
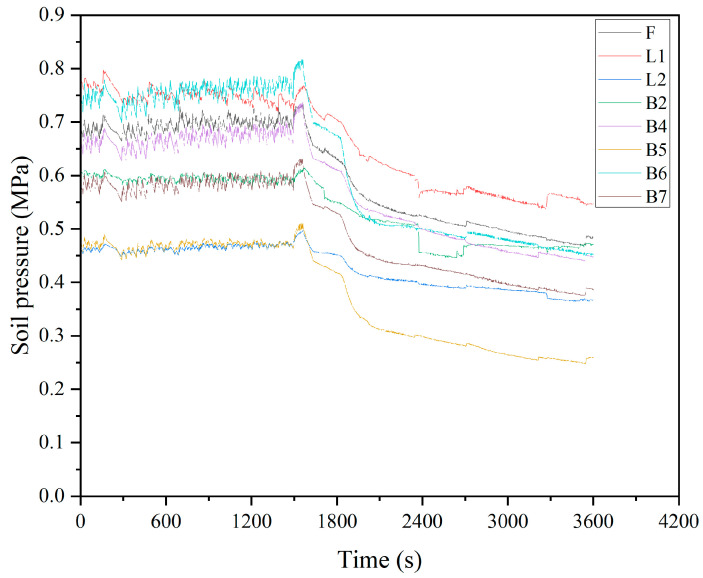
20% missing ratio plot.

**Figure 9 sensors-24-01560-f009:**
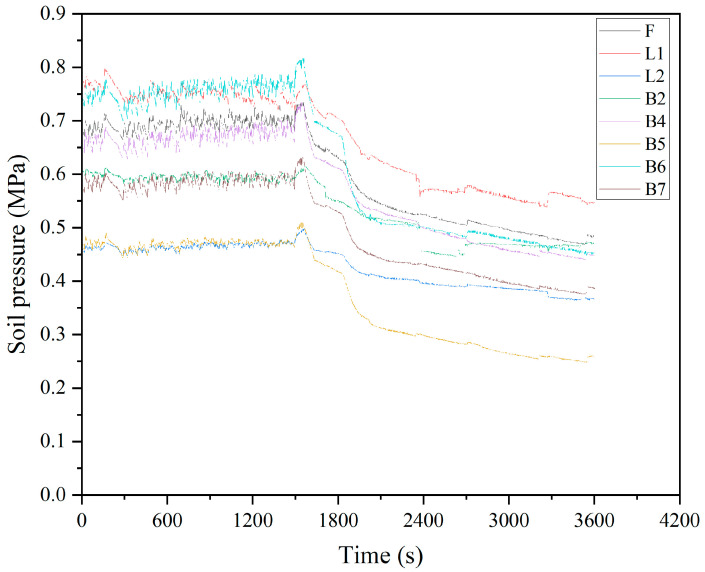
40% missing ratio plot.

**Figure 10 sensors-24-01560-f010:**
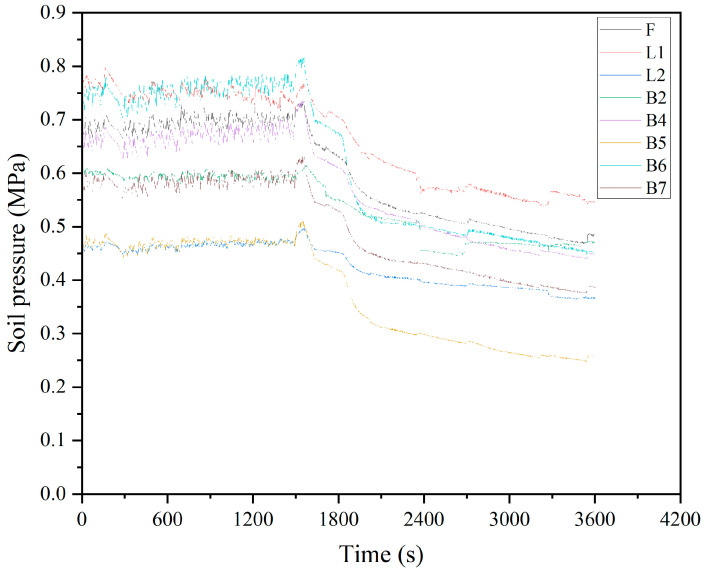
60% missing ratio plot.

**Figure 11 sensors-24-01560-f011:**
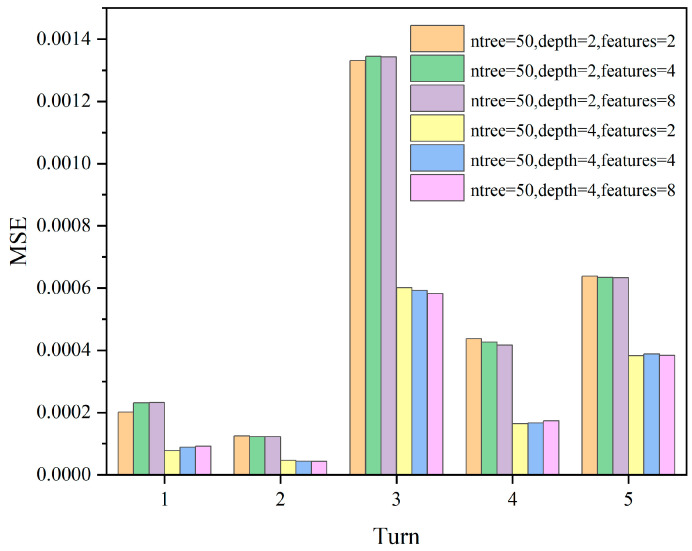
The MSE scores for different parameter combinations of ntree = 50; max depth = 2, 4; max features = 2, 4, 8.

**Figure 12 sensors-24-01560-f012:**
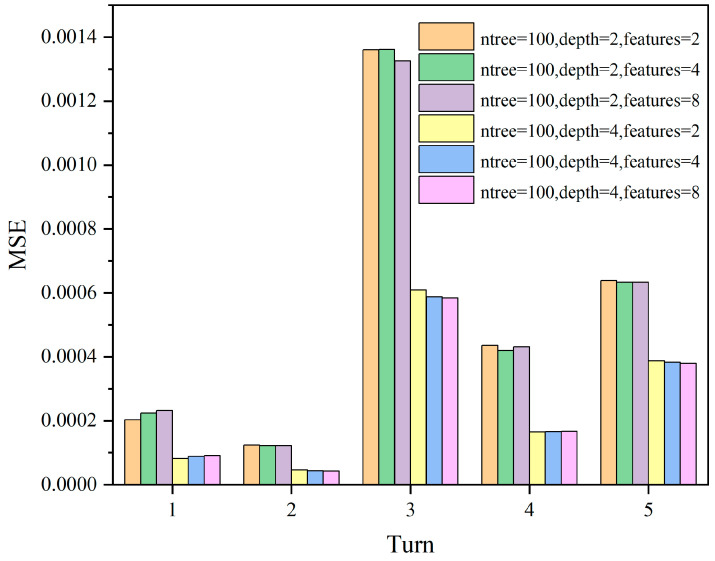
The MSE scores for different parameter combinations of ntree = 100; max depth = 2, 4; max features = 2, 4, 8.

**Figure 13 sensors-24-01560-f013:**
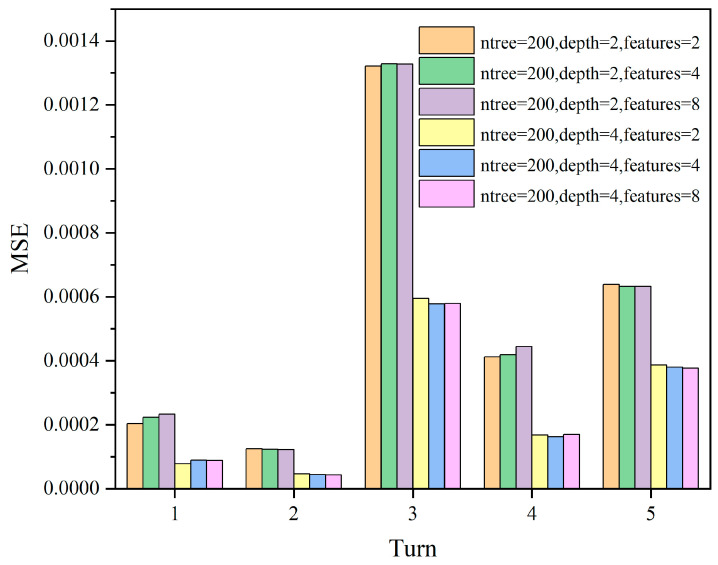
The MSE scores for different parameter combinations of ntree = 200; max depth = 2, 4; max features = 2, 4, 8.

**Figure 14 sensors-24-01560-f014:**
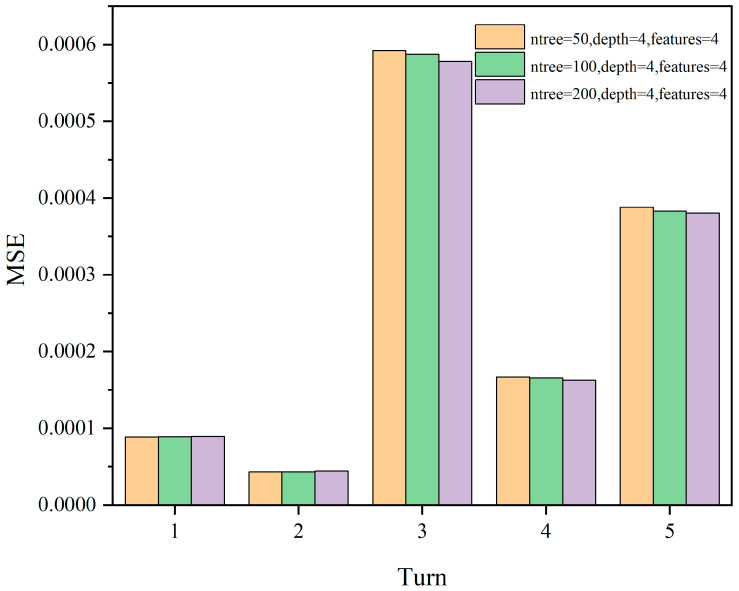
The MSE scores for different parameter combinations of ntree = 50, 100, 200; max depth = 4; max features = 4.

**Figure 15 sensors-24-01560-f015:**
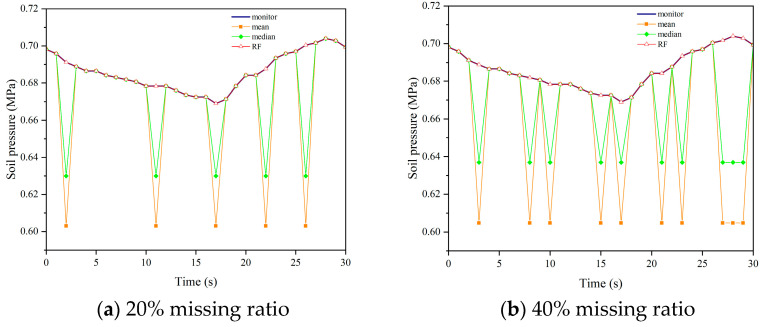
Interpolation results comparison chart of F.

**Figure 16 sensors-24-01560-f016:**
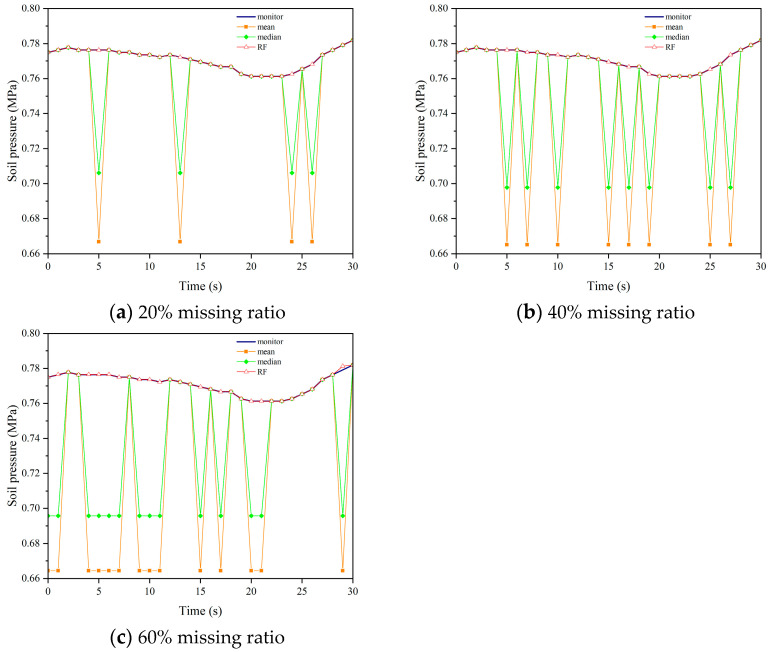
Interpolation results comparison chart of L1.

**Figure 17 sensors-24-01560-f017:**
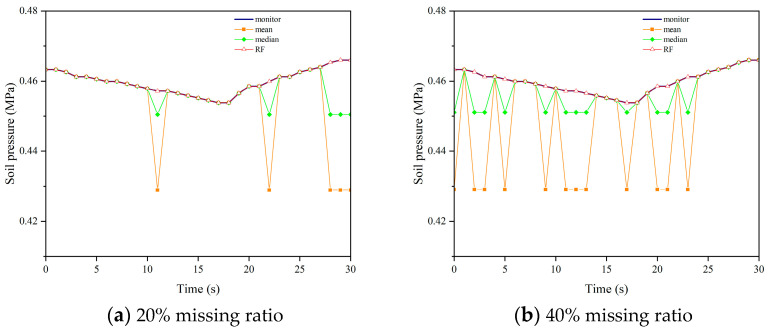
Interpolation results comparison chart of L2.

**Figure 18 sensors-24-01560-f018:**
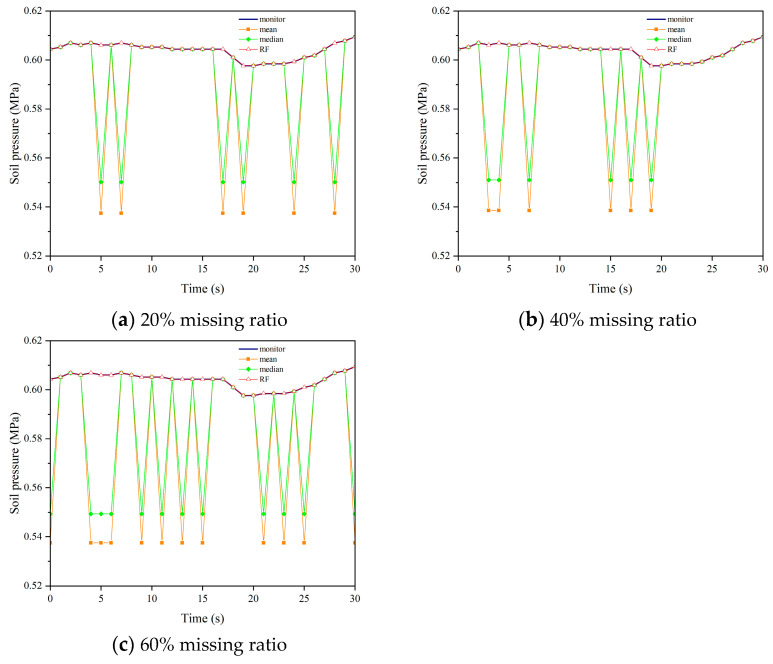
Interpolation results comparison chart of B2.

**Figure 19 sensors-24-01560-f019:**
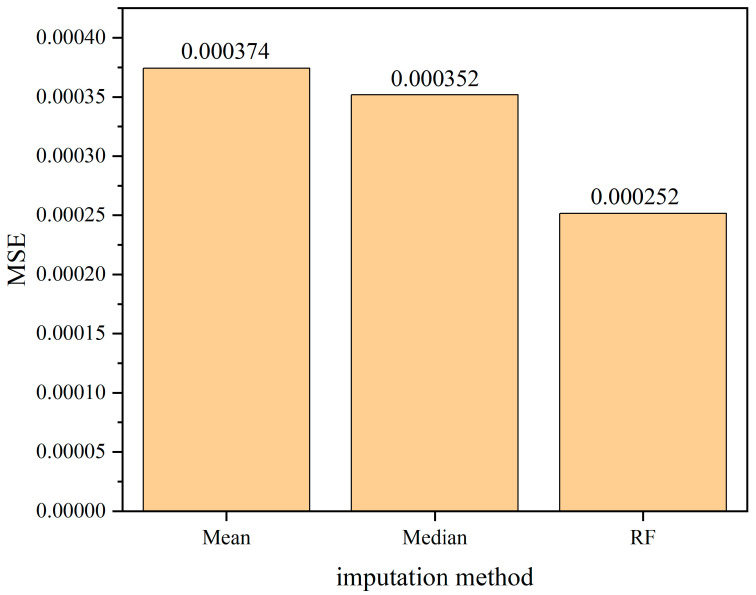
The comparison of mean squared error (MSE) among various interpolation methods under 20% missing data ratio.

**Figure 20 sensors-24-01560-f020:**
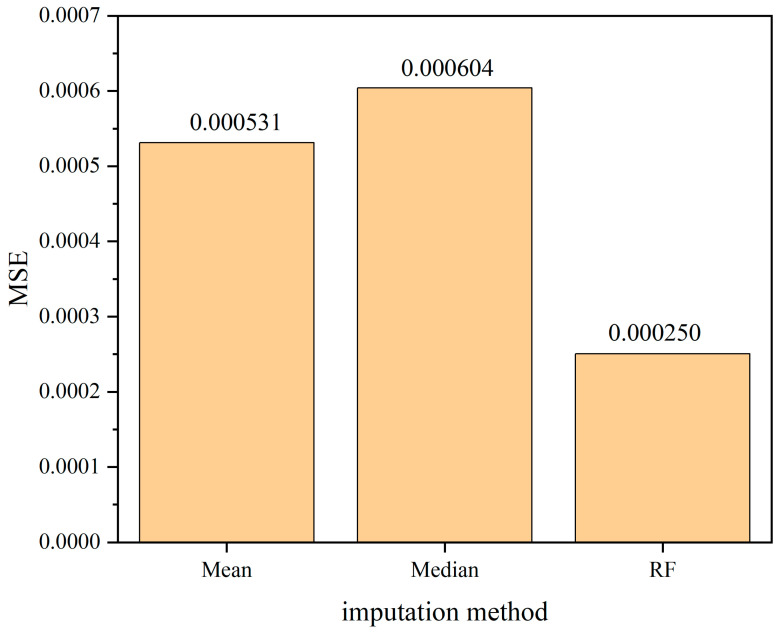
The comparison of mean squared error (MSE) among various interpolation methods under 40% missing data ratio.

**Figure 21 sensors-24-01560-f021:**
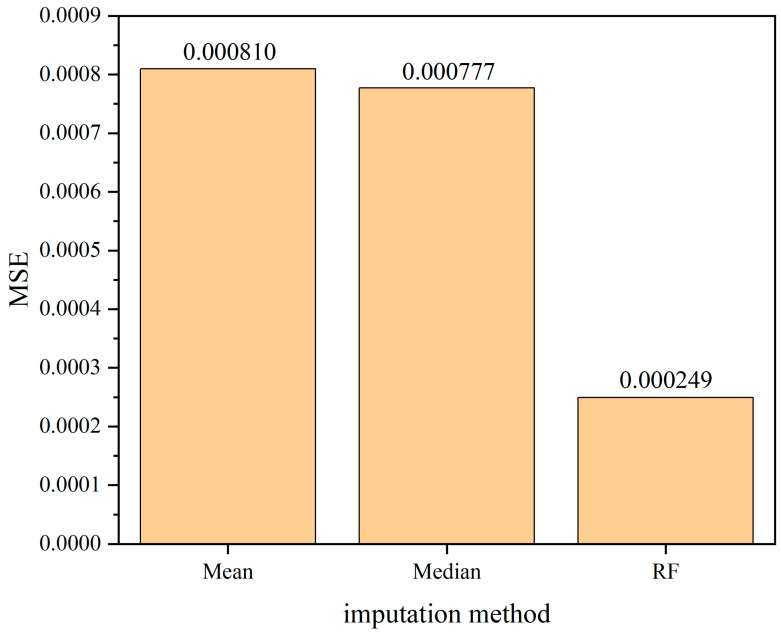
The comparison of mean squared error (MSE) among various interpolation methods under 60% missing data ratio.

## Data Availability

Data are contained within the article.

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
