# Peer review of "Data Imputation of Soil Pressure on Shield Tunnel Lining Based on Random Forest Model"

_sensors, 2024, doi:10.3390/s24051560_

Round 1

Reviewer 1 Report

Comments and Suggestions for Authors

1.      In the abstract, professional terms “MSE” (Line 20) that first appear need to be given its full names.

2.      Line 370, please provide the reason for splitting the dataset into 5 subsets instead of other subsets.

3.      The symbols in formulas (1) - (4) should correspond to the symbols in the explanatory text, including capitalization, italics, and subscripts, etc. The relevant parameters in formula (5) need to be explained.

4.      Line 333, it is suggested providing the additional details for artificially constructing missing datasets with missing ratios of 20%,40% and 60%. as well as a curve chart after providing data with different missing proportions.

5.      It is suggested to provide the curve graph results with different missing ratios after using different imputation methods, to compare it with the original data plot in an intuitive way,

6.      The titles for the vertical axis in Figures 8 to 11 should be consistent with those in Figures 12 to 14.

7.      As the missing proportion increases, the imputation errors of the models based on median and mean imputation methods also increase, while the error of the model based on RF seams to be a regular decrease “0.000252, 0.000250, 0.000249”(Figure12 Figure13 Figure14), please explain it.

8.      It is suggested to incorporate the content of Part 4.1 into Part 1 Introduction, and highlight the main advantages of this method compared to other methods.

9.      The title of the excellent article is about the data imputation method of soil pressure testing data, while in the introduction of tunnel monitoring system in the article, it also mentions monitoring method such as strain and settlement. Why is soil pressure data introduced separately? Is there anything special about soil pressure data? It is suggested to provide additional explanations on the characteristics of soil pressure monitoring data and the reasons for choosing it for analysis.

Author Response

REVIEWER #1:                                                          

  • In the abstract, professional terms “MSE” (Line 20) that first appear need to be given its full names.

Response: “MSE” stands for “Mean Squared Error”, which has been modified in line 20.

  • Line 370, please provide the reason for splitting the dataset into 5 subsets instead of other subsets.

Response: In the random forest algorithm, commonly used cross-validation techniques include 5-fold cross-validation and 10-fold cross-validation, where the dataset is divided into 5 or 10 equal parts. 5-fold cross-validation has become a popular empirical rule in the field of machine learning, validated through numerous experiments. It helps to balance variance and bias when evaluating model performance. We added the following reasons in lines 400 to 404.

  • The symbols in formulas (1) - (4) should correspond to the symbols in the explanatory text, including capitalization, italics, and subscripts, etc. The relevant parameters in formula (5) need to be explained.

Response: The following is a specific description of the symbols and parameters in the formula, and in the text 230 lines to 234 lines, 299 lines to 310 lines, 388 lines to 389 lines have been modified.

 is the soil pressure,  and are the measured and initial wavelengths; and  are the temperature-compensated measured and initial wavelengths;  is the temperature compensation sensitivity coefficient;  is the temperature coefficient; and  is the soil pressure linear coefficient.

In this equation,  stands for the Gini coefficient, where  represents the probabilities of different categories, and  denotes the number of categories.

In this context,  refers to the original dataset,  represents the feature to be partitioned,  denotes the set of samples that satisfy A=a, and  represents the set of samples that do not satisfy A=a.  represents the uncertainty of set , while  represents the uncertainty of set  after it has been partitioned based on A=a.

In this case,  refers to the number of samples,  represents the true values of the test set, and  represents the predicted values of the test set.

  • Line 333, it is suggested providing the additional details for artificially constructing missing datasets with missing ratios of 20%,40% and 60%. as well as a curve chart after providing data with different missing proportions.

Response: Thanks for pointing out this issue. Taking the 20% missing rate as an example, with a total of 28,000 data samples, during the process of artificially creating missing data, we created an array containing 5,760 column indices distributed between 0 and 7, and 5,760 row indices distributed between 0 and 3,599. Subsequently, we used these indices to set the corresponding positions in the dataset to be empty, with the indices randomly generated within the specified range. We added details of artificially creating a missing data set and the curve graphs for different missing data ratios in lines 351 to 363 of the paper.

  • It is suggested to provide the curve graph results with different missing ratios after using different imputation methods, to compare it with the original data plot in an intuitive way.

Response: Thanks for pointing out this issue. We added comparison charts of the results obtained using random forest imputation, median imputation, and mean imputation at different missing data ratios from line 445 to line 471 in the article.

  • The titles for the vertical axis in Figures 8 to 11 should be consistent with those in Figures 12 to 14.

Response: Modifications have been made to figures 11 to 14 in the article.

Figure 11. The MSE scores for different parameter combinations of ntree=50; max depth=2,4; max features=2,4,8

Figure 12. The MSE scores for different parameter combinations of ntree=100; max depth=2,4; max features=2,4,8

Figure 13. The MSE scores for different parameter combinations of ntree=200; max depth=2,4; max features=2,4,8

Figure 14. The MSE scores for different parameter combinations of ntree=50,100,200; max depth=4; max features=4

  • As the missing proportion increases, the imputation errors of the models based on median and mean imputation methods also increase, while the error of the model based on RF seams to be a regular decrease “0.000252, 0.000250, 0.000249” (Figure12 Figure13 Figure14), please explain it.

Response: Median imputation and mean imputation are simple and intuitive methods for filling missing values. As the missing data ratio increases, these methods introduce more errors, leading to an increase in mean squared error (MSE). There is a certain correlation between missing values and the features of existing values. Random forest imputation can capture complex relationships between features, utilize existing data to predict missing values, and obtain more reliable imputed values through the ensemble results of multiple trees. Therefore, as the missing data ratio increases, the MSE of the random forest model stabilizes within a certain range, and there may even be a slight decrease phenomenon.

  • It is suggested to incorporate the content of Part 4.1 into Part 1 Introduction, and highlight the main advantages of this method compared to other methods.

Response: Thank you for your valuable suggestion. We will present this section in lines 105 to 111 of the article.

  • The title of the excellent article is about the data imputation method of soil pressure testing data, while in the introduction of tunnel monitoring system in the article, it also mentions monitoring method such as strain and settlement. Why is soil pressure data introduced separately? Is there anything special about soil pressure data? It is suggested to provide additional explanations on the characteristics of soil pressure monitoring data and the reasons for choosing it for analysis.

Response: Thank you for your valuable suggestion. The soil pressure around the tunnel lining is the primary load sustained during tunnel shield construction and operation. Monitoring data of soil pressure can provide information on stress distribution in the surrounding soil and potential instability, significantly impacting the stability and safety of the tunnel. Moreover, analyzing soil pressure monitoring data allows for identifying deformation characteristics of the lining segments and their relationship with mechanical performance and soil pressure, aiming to optimize tunnel design and construction schemes in the future. We have presented this section from line 214 to line 221 in the article.

Reviewer 2 Report

Comments and Suggestions for Authors

In this paper, the missing data in the data set was imputed using the Random Forest, and its feasibility was verified. The results have reference value for processing missing data.

Questions:

1. Please mark the sensors in Figure 2, such as static water level gauges, vibrating wire strain gauges, soil pressure cells, reinforcement strain gauges, concrete strain gauges, thermometers, displacement meters, and seepage meters.

2. How to consider the authenticity of test results for soil pressure gauges buried in pipe segments?

3. Please provide the data plots of missing rates of 20%, 40%, and 60% corresponding to Figure 7.

4. Generally speaking, after sensor failure, data may be missing or inaccurate. Is the Random Forest based input method to compensate for the data after sensor failure? Why?

5. What does data interpolation mean and where is it interpolated in the data set?

Author Response

In this paper, the missing data in the data set was imputed using the Random Forest, and its feasibility was verified. The results have reference value for processing missing data.

Questions:

  • Please mark the sensors in Figure 2, such as static water level gauges, vibrating wire strain gauges, soil pressure cells, reinforcement strain gauges, concrete strain gauges, thermometers, displacement meters, and seepage meters.

Response: Modifications have been made to Figure 2 in the article.

Figure 2. Sensor schematic diagram

  • How to consider the authenticity of test results for soil pressure gauges buried in pipe segments?

Response: First, the entire monitoring process uses fiber optic grating soil pressure gauges. Fiber optic grating sensors can achieve high-precision detection of small pressure parameters, have good anti-electromagnetic interference and anti-radio frequency interference capabilities, can maintain stable operation in complex situations, have good long-term stability, and can cover relatively long measurement distances, making them suitable for long-distance tunnel monitoring.

Second, the entire installation and pouring process are carried out indoors to ensure that the instruments are not damaged.

Third, before use, we ensure that the soil pressure gauge has been correctly calibrated and verified. During use, we promptly check the working status of the soil pressure gauge to ensure its reliability and accuracy during the measurement process.

Finally, we evenly distribute fiber optic grating sensors around the outer circumference of the entire ring pipe segment. The change trends of data measured by each sensor can be compared with each other to avoid being influenced by local human or environmental factors.

  • Please provide the data plots of missing rates of 20%, 40%, and 60% corresponding to Figure 7.

Response: We have added data plots with missing ratios of 20%, 40%, and 60% at Figure 8 to Figure 10 in the article.

  • Generally speaking, after sensor failure, data may be missing or inaccurate. Is the Random Forest based input method to compensate for the data after sensor failure? Why?

Response: After the sensor failure, no data will be obtained. In this case, alternative prediction methods can be used to forecast short-term data based on the complete dataset before the failure occurred. We are unsure whether the random forest imputation method can predict and fill in the data after the sensor failure.

  • What does data interpolation mean and where is it interpolated in the data set?

Response: Data interpolation refers to filling in missing or incomplete data using existing data. When a dataset has missing data due to various reasons (such as sensor malfunctions or data transmission errors), data interpolation can estimate the missing values using statistical methods or machine learning techniques, thereby repairing the incomplete dataset. Data interpolation occurs at the blank or missing positions in the dataset.

Reviewer 3 Report

Comments and Suggestions for Authors

The article is devoted to the problem of missing data imputation technique. Sensor data of soil pressure on shield tunnel lining can be lost. Imputation of missing data is done on the basis of random Forrest model for time series. It is not obvious that imputation of quasistatic data of sensors allows better estimation of deformation state of tunnel structure. The importance of this technique for filling the time series of sensor data should be demonstrated, and its differences from the time series of, for example, the price of oil barrel.

In general, the results need to be presented more clearly:

- graphs of missing data with different missing ratios should be presented;

- examples of comparing graphs with imputing monitoring data should be presented. The advantages of method should be demonstrated visually.

- What deformation parameter does the MSE value correspond to? (pressure MPa?, strain? ). It is not clear whether MSE = 0.00025 is a big or small value? Perhaps the relative error should also be given.

Also it is necessary to eliminate some shortcoming which are listed in what follows:

- the titles of Figures 8-11 should not contain the text "This figure displays";

- in the same graphs, the name of the Y-axis is CV-score or MSE Score?

The paper can be published after major revision.

Author Response

REVIEWER #3:                                                       

The article is devoted to the problem of missing data imputation technique. Sensor data of soil pressure on shield tunnel lining can be lost. Imputation of missing data is done on the basis of random Forrest model for time series. It is not obvious that imputation of quasistatic data of sensors allows better estimation of deformation state of tunnel structure. The importance of this technique for filling the time series of sensor data should be demonstrated, and its differences from the time series of, for example, the price of oil barrel.

Response: Thanks for pointing out this issue. The incomplete soil pressure dataset makes it difficult to accurately assess the deformation of the tunnel segment structure. A complete soil pressure dataset can more accurately simulate the deformation characteristics and failure modes of tunnel segment structures in three-dimensional finite element simulation software or in the laboratory. We will deepen our research in this area in the future. We have added a section from line 495 to 498 in the paper, emphasizing the importance of using a random forest model-based interpolation method to fill in the missing values in the sensor time series dataset. Additionally, we have included a section from line 102 to 105 highlighting the differences between sensor time series data and other time series datasets.

In general, the results need to be presented more clearly:

- graphs of missing data with different missing ratios should be presented;

Response: We have added data plots with missing ratios of 20%, 40%, and 60% at Figure 8 to Figure 10 in the article.

- examples of comparing graphs with imputing monitoring data should be presented. The advantages of method should be demonstrated visually.

Thanks for pointing out this issue. We added comparison charts of the results obtained using random forest imputation, median imputation, and mean imputation at different missing data ratios from line 445 to line 471 in the article.

- What deformation parameter does the MSE value correspond to? (pressure MPa?, strain? ). It is not clear whether MSE = 0.00025 is a big or small value? Perhaps the relative error should also be given.

MSE stands for Mean Squared Error, which is a metric used to measure prediction model errors. In machine learning and statistics, MSE is a commonly used standard. The range of MSE values is [0, +∞), and in the evaluation process, the closer the MSE is to 0, the smaller the difference between the model’s predicted results and the actual values. In this experiment, an MSE of 0.00025 is a small value.

Also it is necessary to eliminate some shortcoming which are listed in what follows:

- the titles of Figures 8-11 should not contain the text "This figure displays";

- in the same graphs, the name of the Y-axis is CV-score or MSE Score?

Response: Modifications have been made to figures 11 to 14 in the article.

Figure 11. The MSE scores for different parameter combinations of ntree=50; max depth=2,4; max features=2,4,8

Figure 12. The MSE scores for different parameter combinations of ntree=100; max depth=2,4; max features=2,4,8

Figure 13. The MSE scores for different parameter combinations of ntree=200; max depth=2,4; max features=2,4,8

Figure 14. The MSE scores for different parameter combinations of ntree=50,100,200; max depth=4; max features=4

Round 2

Reviewer 1 Report

Comments and Suggestions for Authors

The authors basically address all the comments in the first review of the reviewer. The reviewer is satisfactory with the most replies.

Author Response

Thank you to the reviewer for their support.

Reviewer 2 Report

Comments and Suggestions for Authors

Revised according to review comments and  can be published.

Author Response

(The authors gave the same response as above.)

Reviewer 3 Report

Comments and Suggestions for Authors

After revision of the article it became clear how initial datasets with different missing rates are constructed, and how imputation data obtained by different models visually look like. The data reduction is almost uniform: even data with 60% missing ratio shows the main changes in the sensor readings. It would be much more interesting to see the case where data with significant changes are missing (missing over a longer interval rather than 2-3 points). The authors wrote that "complete soil pressure dataset can more accurately simulate the deformation characteristics and failure modes of tunnel segment structures in three dimensional finite element simulation software". However, this statement requires proof for the considered initial datasets with different missing ratios. For example, use FEM to prove that you lose important information when doing this (I think FEM calculations are not done every second). Or consider other variants of missing data, in which a significant number of data are missing consecutively, on an interval with significant changes.

There is also a question about the correct use of mean and median models in the given examples (Fig.18-21). On the interval shown in the graphs, these models give no logical result at all, worse than using the last known value. Only negative deviations by the same value using these models also do not look logical.

According to Fig. 15-17 MSE random forest model has a difference of 1.5–3 times from the median and mean, but these values are of the same order. However, according to Fig. 18-21 random forrest model is more accurate by 1-2 orders of magnitude. Somewhere in the results there is an error .

The paper can be published after major revision.

Author Response

REVIEWER #3:                                                       

After revision of the article it became clear how initial datasets with different missing rates are constructed, and how imputation data obtained by different models visually look like. The data reduction is almost uniform: even data with 60% missing ratio shows the main changes in the sensor readings. It would be much more interesting to see the case where data with significant changes are missing (missing over a longer interval rather than 2-3 points). The authors wrote that "complete soil pressure dataset can more accurately simulate the deformation characteristics and failure modes of tunnel segment structures in three dimensional finite element simulation software". However, this statement requires proof for the considered initial datasets with different missing ratios. For example, use FEM to prove that you lose important information when doing this (I think FEM calculations are not done every second). Or consider other variants of missing data, in which a significant number of data are missing consecutively, on an interval with significant changes.

Response: Thank you for your valuable input. We have removed the relevant statements from the manuscript, and we will further deepen our research in this area in the future.

There is also a question about the correct use of mean and median models in the given examples (Fig.18-21). On the interval shown in the graphs, these models give no logical result at all, worse than using the last known value. Only negative deviations by the same value using these models also do not look logical.

Response: The median and mean imputation methods fill in missing data based on the median and mean of the entire feature, respectively. They do not consider the feature correlations between missing and existing data. As a result, when these two imputation methods are used to fill in missing values in nonlinear datasets, the filled data may differ significantly from the original data. For example, when using the mean imputation method, the obtained value of “0.602978142259847” can differ by approximately “0.09” from the original data “0.691127306168301.” Similarly, when using the random forest imputation method, the difference between the obtained value “0.691127306168299” and the original data falls within the range of 10-13, highlighting the advantage of the random forest imputation method.

According to Fig. 15-17 MSE random forest model has a difference of 1.5–3 times from the median and mean, but these values are of the same order. However, according to Fig. 18-21 random forrest model is more accurate by 1-2 orders of magnitude. Somewhere in the results there is an error .

Response: Figures 15-18 only demonstrate the comparison between different imputation methods and the original data for missing rates between 0 and 30 seconds. The mean squared error (MSE) measures the average of the squared differences between the actual data and the imputed data. The numerator of the MSE represents the total number of samples in the test set. Therefore, for the final evaluation of the performance and accuracy of the model in imputing missing data, please refer to Figures 19-21. We have added explanatory notes at lines 427-443 in the document, and adjusted the order of the images from Figures 15-21 accordingly.